# A stem group echinoderm from the basal Cambrian of China and the origins of Ambulacraria

Timothy P. Topper [1,2,3], Junfeng Guo[4], Sébastien Clausen [5], Christian B. Skovsted[2] & Zhifei Zhang[1]

Deuterostomes are a morphologically disparate clade, encompassing the chordates (including vertebrates), the hemichordates (the vermiform enteropneusts and the colonial tube-dwelling pterobranchs) and the echinoderms (including starfish). Although deuterostomes are considered monophyletic, the inter-relationships between the three clades remain highly contentious. Here we report, *Yanjiahella biscarpa*, a bilaterally symmetrical, solitary metazoan from the early Cambrian (Fortunian) of China with a characteristic echinoderm-like plated theca, a muscular stalk reminiscent of the hemichordates and a pair of feeding appendages. Our phylogenetic analysis indicates that *Y. biscarpa* is a stem-echinoderm and not only is this species the oldest and most basal echinoderm, but it also predates all known hemichordates, and is among the earliest deuterostomes. This taxon confirms that echinoderms acquired plating before pentaradial symmetry and that their history is rooted in bilateral forms. *Yanjiahella biscarpa* shares morphological similarities with both enteropneusts and echinoderms, indicating that the enteropneust body plan is ancestral within hemichordates.

[1] Shaanxi Key Laboratory of Early Life and Environments, State Key Laboratory of Continental Dynamics and Department of Geology, Northwest University, 710069 Xi'an, China. [2] Department of Palaeobiology, Swedish Museum of Natural History, Box 50007104 05, Stockholm, Sweden. [3] Department of Earth Sciences, Durham University, Durham DH1 3LE, UK. [4] School of Earth Science and Resources, Key Laboratory for the study of Focused Magmatism and Giant Ore Deposits, MLR, Chang'an University, 710054 Xi'an, China. [5] Univ. Lille, CNRS, UMR 8198 – Evo-Eco-Paleo, 59000 Lille, France. Correspondence and requests for materials should be addressed to T.P.T. (email: timothy.topper@nwu.edu.cn)

Echinoderms (pentaradially symmetrical forms) are one of the most familiar and ubiquitous of all animal groups. Because of their sturdy calcitic skeletons, they have a rich fossil record, first appearing during Cambrian Stage 3 (about 520 Ma ago)[1]. However, it was not until the advent of molecular analyses that there was a general agreement as to how echinoderms were related to other deuterostomes[2–11]. Molecular data consistently resolves echinoderms and hemichordates as sister taxa within the clade Ambulacraria, a sister group to the Chordata[2–5]. Yet, the origins and early evolution of the group has been a source of considerable conjecture, not least because the radial body plan of echinoderms is quite different from the bilaterally symmetrical organization of hemichordates and chordates[12]. A pre-radial history of the echinoderms has been suggested, a possibility supported by recent anatomical development analyses[13]. The fossil record has yet to resolve this debate, as the early history of the clade remains poorly documented and the first echinoderms to emerge in Cambrian Stage 3 already appear relatively derived[1,2,12]. Interpretation of characters at the base of the Ambulacraria is further complicated by the apparent divergent morphology of the two hemichordate groups[2] and the body plan of the ancestral ambulacrarian remains idealized.

Here we analyze the systematic affinity and phylogenetic relationships of *Yanjiahella biscarpa*[14] from the Yanjiahe Formation in Hubei, China. Our analysis illustrates that *Y. biscarpa* is a stem-group echinoderm, confirming that plating was acquired by echinoderms prior to pentaradial symmetry and that the early members of the clade were bilaterally symmetrical. *Yanjiahella biscarpa* exhibits morphological characters that are shared with both enteropneusts and echinoderms, demonstrating that the enteropneust body plan is ancestral within the Hemichordata.

The mosaic of morphologies that this taxon displays may illustrate the deepest roots of the Ambulacraria and is therefore crucial for interpreting the early history of the deuterostome clade.

## Results and discussion
### Systematic paleontology.

<div align="center">

Total-group Echinodermata

Class, Order, Family indet.

*Yanjiahella biscarpa* Guo, Li, Han, Ou, Zhou and Zheng (2012).

*Yanjiahella biscarpa*; Guo et al. (2012), p. 793, Figs. 2k–m, 3c.

*Yanjiahella ancarpa*; Guo et al. (2012), p. 793, Figs. 2a–c, 3a.

*Yanjiahella monocarpa*; Guo et al. (2012), p. 795, Figs. 2d–j, 3b.

</div>

**Holotype**. ELI-HS722 (see Guo et al.[14], Fig. 2k). Originally three species were nominally described in the same publication[14], based on the number of preserved "arms" (here termed feeding appendages). However, it is clear that the three species share the same morphological features, including thecal and stalk characters and the number of appendages is simply a taphonomic artifact. We here, as the First Reviser, consider the three species to be subjective synonyms, and under Art. 24.2.2 of the International Code of Zoological Nomenclature (ICZN), we give precedence to the species name *Y. biscarpa* (referring to the possession of two feeding appendages).

**Material**. Thirty five specimens.

**Locality**. Gunziao Section, Sandouping, Yichang, Hubei province, China (Supplementary Figure 1).

**Stratigraphy**. Specimens recovered from gray-black, silty shale intercalations in a carbonaceous limestone of the Yanjiahe Formation[14]. This part of the Yanijahe Formation is

known as Bed III[14] and is Fortunian in age[14–16] having been correlated with the *Purella antiqua* shelly fossil zone[14] (Supplementary Figure 1). Carbon isotope chemostratigraphic studies support this age[15,16].

**Description**. Small (20–50 mm in height) bilateral animal having a globular to conical multi-plated theca, two long organic walled feeding appendages, and a stalk divided into two morphologically distinct zones (proximal and distal). The theca is covered with numerous irregular, ovoid to polygonal plates (Figs. 1a–c, 2a, c and Supplementary Figs. 2a, 3a, d, g). Plates are up to 2.5 mm in length and width with an unknown original composition. In some specimens, partial recovery between adjacent plates suggests that the theca might have been imbricately plated (Fig. 2a and Supplementary Fig. 3a, c). However, plates seem to be grossly abutted in other specimens or areas, so that definite assumption over plating type would be overstated (Fig. 1a and Supplementary Figs. 2a, 4b, d, h, 5a). Thecal plating is often preserved as slightly dislocated (a possible alternative explanation for apparent imbrication of plates) with juxtaposition of internal and external molds. Due to the variation in outer shape and size of individual plates, plating was most probably irregularly arranged during life, with plates loosely embedded within tegument. The variable shape of the theca (Figs. 1a–c, 2a and Supplementary Figs. 2a, 3a, d, 4a–d, g–h, 5a, e, i) also supports this deduction. Some specimens show an almost continuous transition between stalk and theca (Fig. 2f, g and Supplementary Figs. 4a–d, 5a, e, i), so only the change in plating marks that boundary between these areas, with plates being absent in the stalk. Thecal plates have differentiated smooth (inner) surface and an outer (external) surface ornament of parallel ribs (Fig. 1c, 2a and Supplementary Figs. 3a, c). The proximal stalk is cylindrical (Figs. 1a, d, 2a, f, g and Supplementary Figs. 2b–c, h, 3a, e, 4f, i, 5e, i) and tapers distally (Fig. 1a, e and Supplementary Figs. 2c, h, 4g, 5a, e, i). The proximal stalk is transversely ridged (distance between two successive ridges about 100 μm); the ridges are usually very fine, but can be locally thickened (Fig. 1c, d, Supplementary Fig. 2b). The distal stalk is not transversely ridged but finely, longitudinally striated (Fig. 1e, g and Supplementary Figs. 2e–h, 5h) and is otherwise in perfect continuity with the proximal stalk. The relative rigidity of the stalk seems to decrease distally as the proximal part is generally almost straight with accentuated folds (Figs. 1, 2 and Supplementary Figs. 2, 3, 5a, e, i), although it is sometimes gently curved distally (Fig. 1 and Supplementary Fig. 2), while the striated distal zone of the stalk is invariably curved and looped (Figs. 1a, 2a and Supplementary Figs. 2a, 3a, 4g, 5a, e, i). A median, internal structure, visible from the posterior margin of the theca extends through the muscular stalk (Fig. 2a, d, f and Supplementary Figs. 5a–d). This structure is interpreted as the digestive tract and it continues through the proximal stalk region into the distal zone of the stalk (Fig. 2d and Supplementary Figs. 4g, 5a–d). However, the digestive tract always disappears before the posterior extremity of the distal stalk (Fig. 2d and Supplementary Figs. 4g, 5a–d). This region of the distal stalk that lacks the digestive tract is interpreted as a post-anal anchoring organ. The digestive tract is not always discernible in the upper region of the distal stalk (Fig. 1e and Supplementary Fig 2), possibly due to taphonomic factors, but its detection may also be masked by the strong, parallel longitudinal striations of this part of the distal stalk (Fig. 1e and Supplementary Figs. 2e, g–j). Two appendages project from opposing lateral sides of the thecal margin. The appendages are long and slender and can extend up to 43 mm in length and may reach 0.6 mm in diameter. The appendages are not covered by plates, but sometimes transverse lines are observable, which could represent cases when two appendages are preserved juxtaposed to each other (Fig. 2a and Supplementary

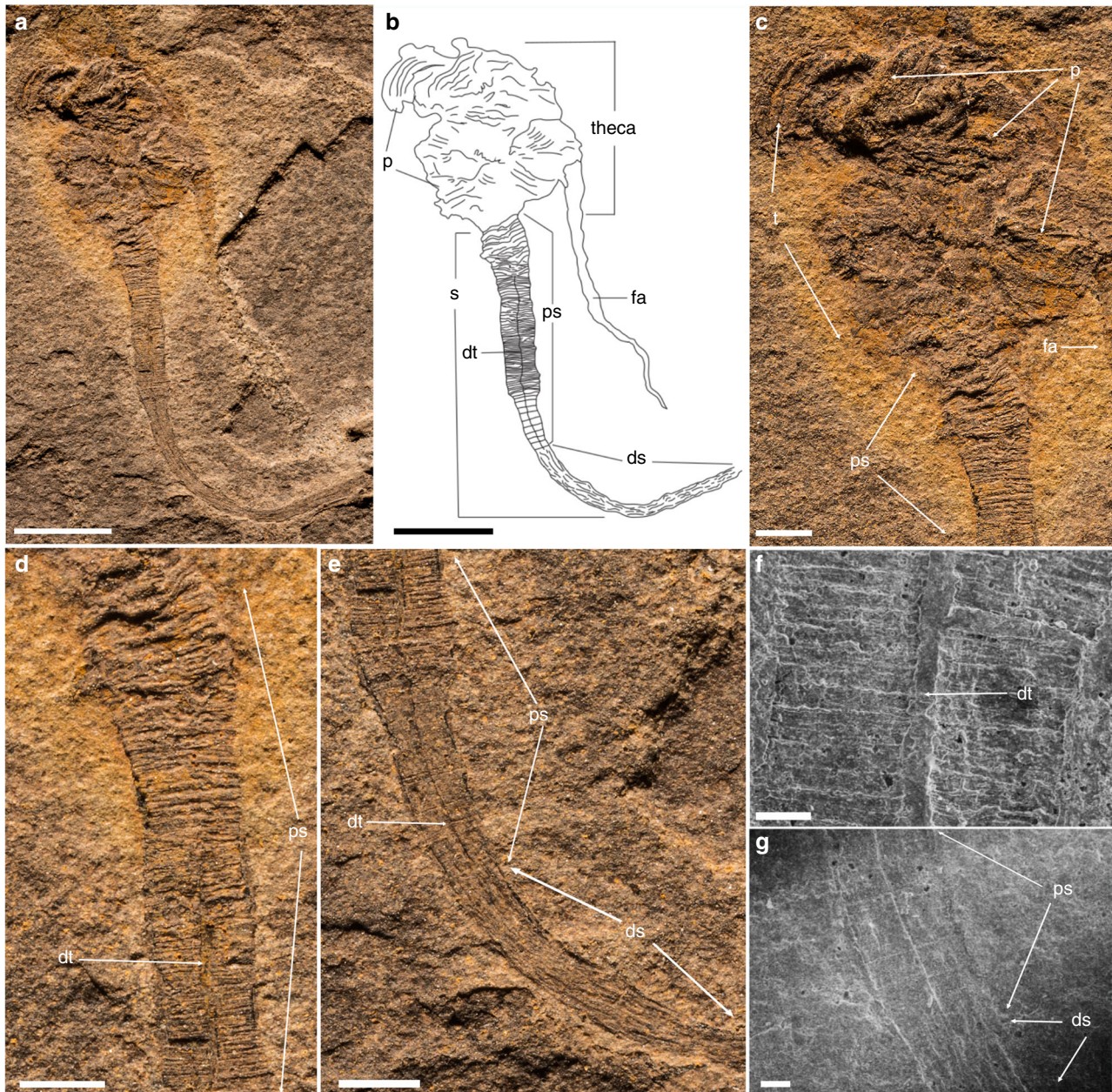

**Fig. 1** *Yanjiahella biscarpa* from the lower Cambrian (Fortunian) Yanjiahe Formation in China. Specimen ELI-HS18. **a** Specimen with plated theca, a muscular stalk, and a single feeding appendage. **b** Line drawing of specimen. **c** Detail of globular theca with plates, a feeding appendage and proximal stalk with distinct transverse ridges. **d** Detail of proximal stalk showing transverse ridges and a digestive tract. **e** Details of the stalk showing transition from the transverse ridged proximal stalk to the striated distal stalk. **f** Scanning electron microscopy (SEM) image of the digestive tract, with transverse ridges visible. **g** SEM image of the transition from the proximal to distal stalk. Scale bars, 5 mm (**a**, **b**), 1 mm (**c**–**e**), and 200 μm (**f**, **g**). ds, distal stalk; dt, digestive tract; fa, feeding appendage; p, plate; ps, proximal stalk; s, muscular stalk; t, theca

Fig. 3c). The variable shape of the theca, the strong curvature of the distal stalk and the incomplete preservation of some specimens has hindered any reliable quantitative study. Consequently, details regarding the ontogeny of *Y. biscarpa* are unknown. No body openings (mouth, gill slits, anus) have been identified in any of the specimens.

**Morphological characteristics.** *Yanjiahella biscarpa* exhibits a globular theca consisting of irregularly arranged, ovoid to polygonal plates (Figs. 1a–c, 2a–c and Supplementary Figs. 2a, 3a, d–g, 4d, e, h). Polygonal to irregular ovoid plates of a similar size (up to ~2.5 mm in length) and arrangement are common in early

echinoderms[17] and thecal plating is generally taken as diagnostic for the phylum[2,7,12]. However, *Y. biscarpa* is unique in the parallel-ridged ornamentation of its thecal plates (Figs. 1a–c, 2a–c and Supplementary Figs. 2a, 3a, d, g). The theca of *Y. biscarpa* is constructed of poorly articulated plates most probably loosely embedded within tegument, a common feature among early echinoderms, as displayed in ctenocystoid and cinctan teguments[18], gogiid theca, and imbricated eocrinoids[19]. No trace of the unique echinoderm stereom mictrostructure[20] or characters such as mouth, gonopores, or possible gill slits have been observed in *Y. biscarpa*. However, this cannot be taken as an indication that these particular echinoderm apomorphies are absent, as stereom microstructure and thecal body openings are

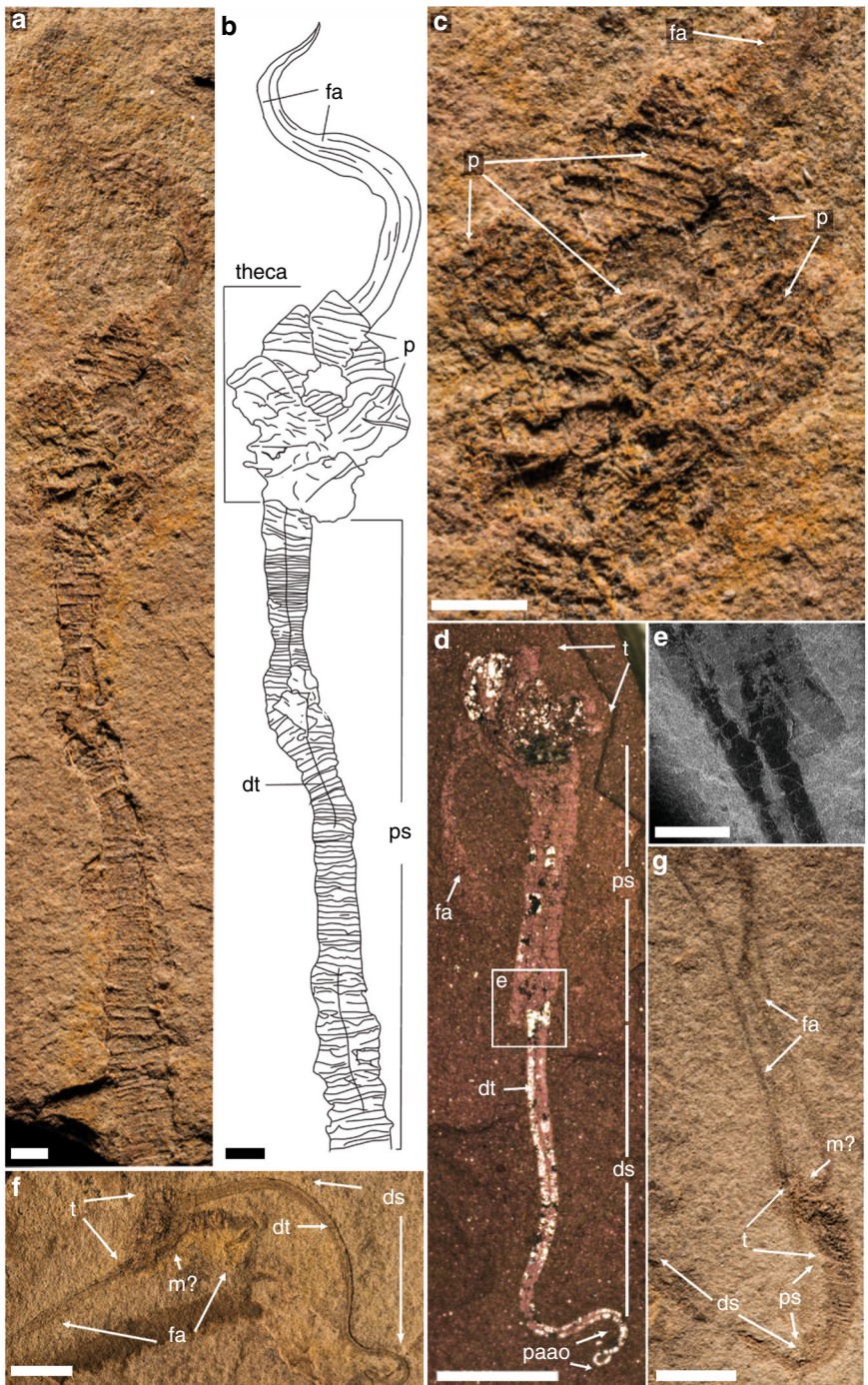

**Fig. 2** *Yanjiahella biscarpa* from the lower Cambrian Yanjiahe Formation in China. Specimen ELI-HS84 (**a–c**). **a** Specimen with plated theca, a muscular stalk and feeding appendages. **b** Line drawing of specimen. **c** Detail of theca, and ornamented plates. Specimen G21 (**d**, part and **e**, counterpart). **d** Specimen with theca, stalk, and a feeding appendage. **e** Image of ruptured stalk. **f** Specimen ELI-HS998B, Specimen with two feeding appendages, and a muscular stalk showing longitudinal folds. Specimen ELI-HS999 (**e**, **f**). **e** Specimen with slightly distorted theca, a stalk and feeding appendages. **f**. Detail of stalk with digestive tract. Scale bars 1 mm (**a–c**, **e**), 2 mm (**e**), and 5 mm (**d**, **g**). ds, distal stalk; dt, digestive tract; fa, feeding appendage; m, mouth; p, plate; ps, proximal stalk; paao, post-anal anchoring organ; s, muscular stalk; t, theca

generally not observed in loosely plated echinoderms preserved as compressed molds in siliciclastic sediments[12,19,21].

The theca of *Y. biscarpa* is prolonged by a muscular stalk that is always preserved as a flattened external mold. The proximal and distal zones of the muscular stalk are quite distinct. The proximal stalk is always preserved as a straight to gently curved structure with transverse ridges (Figs. 1, 2a, b) and was most likely more

rigid than the distal stalk, which is invariably curved and looped and may have been organic (Fig. 2d, f). The distinct morphological differences and contrasting rigidity between the two regions of the stalk can be explained by the presence of a lightly mineralized, tubular exoskeleton that surrounds the proximal region of the muscular stalk. The minute, yet distinctive transverse ridges that are confined to the proximal stalk

(Figs. 1a–e, 2a–e and Supplementary Figs. 2a–d, 3a, e, 4c–g, 5g–l) would represent the exterior of the lightly mineralized tubular exoskeleton. The absence of these transverse ridges in the distal stalk, together with the apparent increased flexibility and the presence of longitudinal striations (which are suggestive of musculature), indicates that no exoskeleton is present in this region.

In one specimen, the wall of the proximal stalk is irregularly torn (Fig. 2d, e). This rupture crosses transverse ridges, but does not truncate the soft part of the stalk (Fig. 2d, e). This supports our interpretation that the wall of the proximal stalk was lightly mineralized and significantly indicates that the stalk of *Y. biscarpa* was not composed of articulated columnal elements or plates. In addition, this specimen illustrates that the stalk was devoid of an endoskeleton, which could not break without affecting the surrounding soft part of the stalk. Also, no disarticulated elements or incomplete disarticulation of the stalk has been observed. As such, the proximal stalk of *Y. biscarpa*, which is associated with a tubular external skeleton, is novel amongst echinoderms, and is very different from the multi-plated stalk or posterior appendage of derived Cambrian representatives of the group[1,22]. The ability to secrete tubular structures is instead a character reminiscent of extant and fossil hemichordates[21–25]. Pterobranchs secrete and live in a tough, external tubarium[26,27], a character that is regarded as homologous to the tubes of the Cambrian enteropneusts *Spartobranchus tenius* and *Oesia disjuncta*[23,25]. Similar to the proximal stalk of *Y. biscarpa*, the tubular constructions of *Spartobranchus tenuis* typically have a ribbed appearance and are prone to tearing[23]. This is not to say that as *Y. biscarpa* was tube dwelling, with the ability to retract or leave this skeleton, like pterobranchs and the Cambrian enteropneusts[25,26,28], but rather that the weakly mineralized, cylindrical exoskeleton most likely provided additional structural support for the organism. Although the tubes of *S. tenius* and *O. disjuncta* are not considered homologous to the mucuous tubes of living enteropneusts[29], the ability to construct a tubular external structure is ubiquitous within the hemichordate clade and no equivalent is known in echinoderms.

The median axis of the stalk of *Y. biscarpa* is marked by a linear, sometimes dark structure that extends posteriorly from its insertion within the theca, terminating before the distal extremity of the stalk (Figs. 1, 2 and Supplementary Figs. 2–5). This linear structure extends beyond the proximal stalk into the non-skeletonized distal, muscular stalk so it cannot represent linear articulation or ornamentation within the proximal exoskeleton (see also discussion above regarding absence of articulated elements). On the contrary, for the most part, this structure is preserved as a flattened lineation, even when the wall ridges of the stalk are preserved in relief, suggesting this structure was organic (Figs. 1a–f, 2a–b, d, f and Supplementary Figs. 2c, 3a, e, 4e–f, h–i, 5a–d). In colonies of pterobranchs (e.g., *Rhabdopleura*), a stolon connects individual zooids within the colony[27]. However, the stolon in pterobranchs is laterally embedded in the secreted wall[27], a position never observed in specimens of *Y. biscarpa*. Specimens of *Y. biscarpa* are also always found as isolated individuals, providing no support that this structure would function in the same way as a stolon. Enteropneusts have a median digestive tube running from the mouth to the anus, with the anus located either at the end of the muscular tail, or just before post-anal anchoring tail in harrimanid larvae[9,30]. The posterior end of the muscular stalk of *Y. biscarpa* also lacks this median structure, indicating that the digestive tract terminated before the distal extremity of the muscular stalk and this region is interpreted as a post-anal anchoring organ, similar to the enteropneust anchoring tail (Fig. 2d, f and Supplementary Figs. 4g, 5a–l). The muscular stalk of *Y. biscarpa* is therefore

here interpreted as a muscular stalk reminiscent of enteropneusts with a weakly mineralized, cylindrical skeleton surrounding its proximal part, with a distal post-anal anchoring organ and the median linear structure itself being a digestive tract.

Opposite the stalk, two elongate appendages extend from the theca (Fig. 2f, g and Supplementary Figs. 4a–d, 5). Their preservation does not allow a detailed description, but the structures are invariably curved and lack any supporting mineralized ossicles. They were originally soft and flexible in life, a sinuosity emphasized by their remarkable length (up to 43 mm in length; appendage/body length ratio about 2.5, maximum 3.3). Similar high feeding appendage/body length ratios have also been observed in early pentaradiate filter-feeding blastozoans (e.g., gogiids[31]), but Cambrian echinoderm feeding appendages are always supported by mineralized ossicles. They also differ completely in morphology to the complex feeding tentacles of pterobranchs[7]. On some specimens, the two appendages project from opposing lateral sides of the theca, imprinting a distinct bilateral symmetry to the animal body (Fig. 2f, g and Supplementary Figs. 4a, b, 5). The projecting appendages are here interpreted as soft feeding organs, the precise nature of which (muscular, collagenous, and or hydrovascular) cannot be assessed. Although it is not preserved in any specimens, the mouth was most likely to have been positioned on the margin of the theca between the bases of the two feeding appendages (Fig. 2f, g and Supplementary Fig. 5c, f, j).

**Phylogenetic significance.** *Yanjiahella biscarpa* was a bilaterally symmetrical animal that exhibits a combination of echinoderm and hemichordate features. On the basis of possessing a plated theca, *Y. biscarpa* is here considered to be a stem echinoderm, basal to the symmetrical to weakly asymmetrical ctenocystoids and other asymmetrical cinctans. To test this hypothesis and evaluate the evolutionary implications of our observations, we designed a phylogenetic analysis of 21 taxa (20 ambulacrarian taxa with cephalochordates used as an outgroup taxon), each scored for 42 morphological characters (Supplementary Note 1 and Supplementary Table 1). Whether the analysis is performed under Traditional Search, New Technology Search, or Implicit Enumeration[32] and despite low bootstrap support, *Y. biscarpa* is constantly resolved as a basal echinoderm (Fig. 3 and Supplementary Figs. 6, 8, 9), reflecting our proposal that it represents a stem-group echinoderm. Only majority-rule consensus trees obtained under Traditional Search methods show *Yanjiahella* in an alternative position, as a basal hemichordate (50% majority-rule consensus tree, Supplementary Fig. 7a) or in a polytomy with ctenosytoids and cinctans as basal ambulacrarians, together with the hemichordate and remaining echinoderm clades (75% majority-rule consensus tree, Supplementary Fig. 7b). These rare deviations most likely highlight the morphological disparity of taxa at the base of the ambulacrarian clade.

In the majority of analyses, *Y. biscarpa* either occupies a basal position within the monophyletic total-group Echinodermata (Fig. 3 and Supplementary Figs. 6b, 8b) or is grouped with ctenocystoids as the sister group to the remaining echinoderms (Supplementary Figs. 6a, 8a). In the latter cases though, the grouping of *Y. biscarpa* and ctenosystoids is never supported by any synapomorphies and is most likely a result of both taxa lacking characters present in more derived echinoderms. Indeed, the basal position occupied by *Y. biscarpa* within the Echinodermata is most likely a result of three factors: (1) the presence of a plated theca with possible stereom microstructure (although such a presence was questioned in our analysis as it is not definitely demonstrated); (2) the absence of blastozoan-grade characters, such as ambulacra, plated stalk, and skeletonized

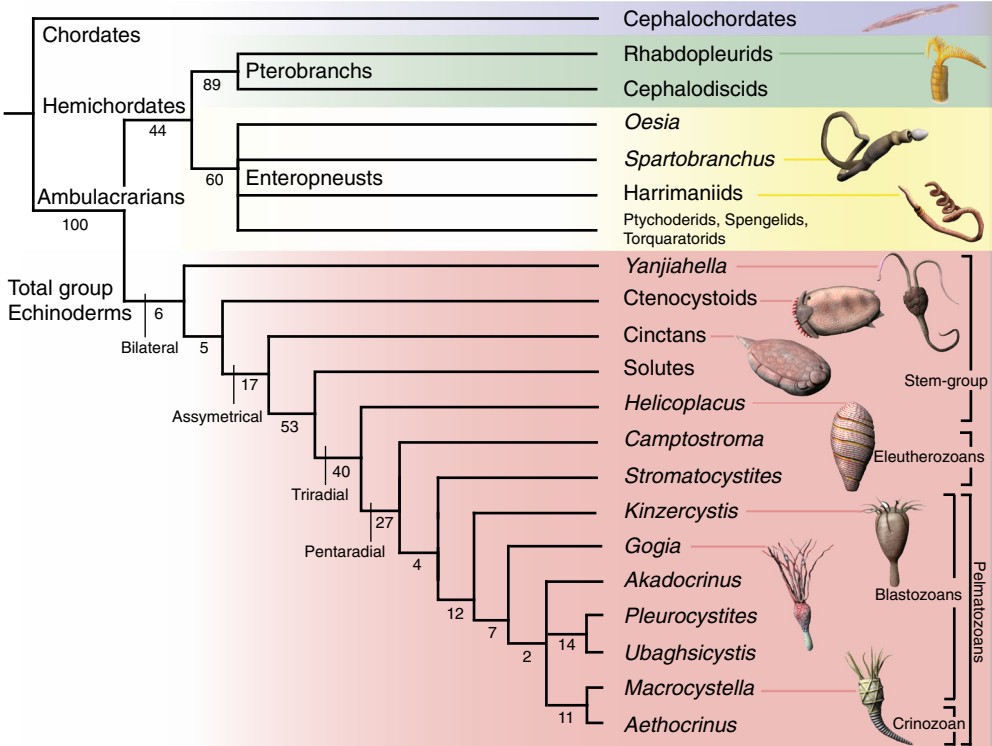

**Fig. 3** Phylogenetic position of *Yanjiahella biscarpa* within the Ambulacraria. Topology based on implicit enumeration with autocollapse search off and resampling via standard bootstrap methods. Bootstrap values shown under each node. Fifteen trees were found with a total length of 71 steps (Consistency Index = 0.648, Rentention Index = 0.821). Taxa illustrated by Nobumichi Tamura

feeding appendages; and (3) the persistence of a number of features akin to the hemichordates, like the presence of an antero-posterior axis and possession of a muscular stalk.

This analysis confirms that echinoderms acquired a skeleton before pentaradial symmetry and that the early history of the Echinodermata is firmly rooted in bilateral forms. This topology is largely consistent with previously published trees dealing with Cambrian echinoderms[33,34] (despite significant changes in the data matrices), indicating pentaradial symmetry evolved progressively from bilateral, asymmetrical, and triradial forms (Fig. 3). The presence of a posterior muscular appendage in *Y. biscarpa* confirms that the posterior appendage and its muscular component evolved numerous times among echinoderm lineages. Other stem-group echinoderms such as ctenocystoids lack a posterior appendage[34,35], yet the posterior appendage of more derived taxa such as solute, stylophoran, pleurocystiid blastozoans, and some coiled stem crinoids[36] have been interpreted as being muscular[1,2,21,22,37–39]. Previous studies have also hinted at this convergence, with the suggested subdivision of blastozoans and crinozoans within the pelmatozoan echinoderms based on the independent acquisition of the stem in crinoids and blastozoans[31,40]. The muscular stalk of *Y. biscarpa*, however, is morphologically disparate to the articulated plated stalks or posterior appendages of derived Cambrian echinoderms. The muscular appendage of the latter is always plated and never contains the digestive tract, which is enclosed within the theca. All echinoderms except ctenocystoids have lost their antero-posterior axis[37,41], Cambrian forms having a lateral or upper anus opening through the thecal wall, next to the mouth as a result of torsion during metamorphosis[1,12,41].

The weakly mineralized, ridged cylindrical exoskeleton surrounding the proximal region of the stalk of *Y. biscarpa* also has no obvious analogy in echinoderms and instead resembles the flexible tubular external constructions secreted by some members of the Hemichordata. The secretion of tubular external skeletons has long been considered to occur only in pterobranchs; however, recent fossil discoveries of enteropneusts from the Cambrian[23,25] have demonstrated that the construction of such external tubular structures are present in both clades. In our analysis, the ability to secrete an external tubular structure resulted in an unresolved phylogenetic signal within the hemichordate clade (Fig. 3). Although our phylogenetic analyses always recognized the enteropneusts and pterobranchs as belonging to two distinct, sister hemichordate clades (as expected), the exact relationships between the enteropneust taxa was less clear. The Cambrian enteropneusts were either considered as derived (with the crown group families as basal; Supplementary Fig. 6a) or basal to the extant enteropneusts (Supplementary Fig. 8a). This phylogenetic uncertainty is highlighted with bootstrap resampling as the group is always unresolved (Fig. 3 and Supplementary Figs. 6b and 8b). This ambiguity is most likely a result of insufficient morphological data to resolve meaningful relationships among hemichordate taxa, in particular the position of the Cambrian taxa, *S. tenius* and *O. disjuncta*.

The antero-posterior axis and the muscular stalk surrounded by a tubular structure in *Y. biscarpa* are reminiscent of the hemichordates, especially the enteropneusts. Although both hemichordate groups have the ability to build tubular structures, the exoskeleton surrounding the proximal stalk of *Y. biscarpa* lacks a stolon system, and does not exhibit the unique zig-zag, fusellar structure characteristic of most pterobranchs[26], providing little evidence linking this structure to the pterobranch coenecium. In addition, while pterobranchs have a typical u-shaped gut[10,41], *Y. biscarpa* exhibits a linear digestive tract that extends through the majority of its body, a configuration more reminiscent of enteropneusts. We therefore follow herein

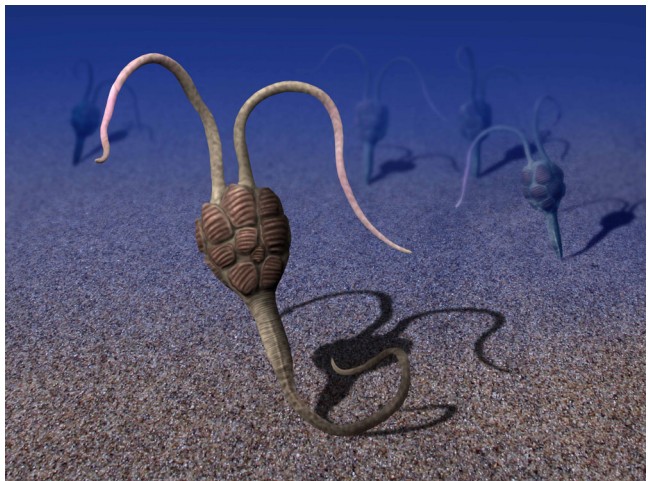

**Fig. 4** Reconstruction of *Yanjiahella biscarpa* on the Cambrian seafloor. The precise arrangement of the plates on the theca is unknown. Artwork by Nobumichi Tamura

previous phylogenetic studies[23,25,29] in interpreting the tubicolous habit as an ancestral trait of the Enteropneusta that was ultimately lost in the crown group enteropneusts (Fig. 3).

In addition to its antero-posterior body axis, morphological features of the stalk of *Y. biscarpa*, such as its muscular nature, its post-anal anchoring organ, and the presence of a linear digestive tract, were most probably inherited from the last common ambulacrarian ancestor, suggesting that the enteropneust body plan is more basal among hemichordates. However, the acquisition of plates embedded in the body wall, a synapomorphic character that unites all total-group echinoderms, firmly positions *Y. biscarpa* as the basalmost known echinoderm. As a consequence, the muscular hemichordate-like stalk, tubular exoskeleton, and post-anal anchoring organ must have regressed before the emergence of the still bilateral ctenostyoids. In life, *Y. biscarpa* was most likely a suspension feeder, standing erect above the seafloor (Fig. 4), or possibly a deposit feeder, lying recumbent on the seafloor. In both cases, the distal portion of the tail was most probably an anchoring organ, comparable to the post-anal tail of some juvenile enteropneusts. The post-anal anchoring organ may have been inserted within the uppermost layer of the sediment, with the anus positioned just above the seafloor (a scenario illustrated in Fig. 4). Alternatively, by analogy with some juvenile enteropneusts[9,30], the complete distal stalk may have been inserted within a short vertical burrow. Although the distal stalk of *Y. biscarpa* was relatively long, none of the observed specimens suggests that it was able to produce complete U-shaped burrows. Such a configuration would have allowed adjustment of its erected portion above the seafloor (variable tiering) by anchoring the post-anal organ at different levels of the burrow wall. To conclude, whatever the favored interpretation, *Y. biscarpa* was most probably a motile benthic, basalmost echinoderm, able to use its muscular stalk to move over the seafloor before anchoring in feeding position.

Despite continual support for the monophyly of the deuterostomes[5,11,42], the instability in hemichordate phylogeny[7,28,42] dramatically affects the reconstruction of ancestral echinoderms and impedes our understanding of the early stages of deuterostome evolution. This discovery has opened new possibilities in the quest for elucidating the early evolution of the Echinodermata, as *Y. biscarpa* predates all known Cambrian representatives by several million years[1], and is among the earliest reported deuterostomes[42].

The body plan of the last common deuterostome ancestor has been hotly debated with the enteropneust or pterobranch arrangement as competing models[2,7,23,43]. As the oldest total-group echinoderm known to date, *Y. biscarpa* illustrates the early divergence of echinoderms from an enteropneust-like ambulacrarian ancestor. Consequently, our study dismisses the pterobranch-like body plan that was traditionally assumed to better illustrate the deuterostome ancestor[44,45].

## Methods

**Material studied**. The fossil material studied herein is deposited at the Early Life Institute of Northwest University, Xi'an (Prefix: ELI). A few specimens were mechanically prepared using a tungsten-tipped micro-engraving tool.

**Phylogenetic analysis**. The phylogenetic analysis was conducted using TNT1.1[32] under implicit enumeration (Fig. 3), Traditional Search with 10,000 replicates (Supplementary Figs. 6, 7), and New Technology Search, using Driven Search with Sectorial Search and Tree fusing options, and the analysis was set to find the minimum tree length 100 times (Supplementary Figs. 8, 9). In all analyses, all characters were treated as unordered and with equal weighting. Bootstrap analyses were also completed on each tree to assess the reliability of individual nodes on the tree. Please see Supplementary Note 1 for further details.

**Image acquisition**. Specimens were photographed using both a Canon EOS 5D Mark III and a stereophotographic Zeiss Smart Zoom 5. Some images were taken when the specimens were coated with ammonium chloride to enhance contrast, and others were imaged using low vacuum scanning electron microscopy. Figures were constructed using Adobe Photoshop and Illustrator CS6 and any measurements were taken using ImageJ.

**Nomenclatural acts**. This published work has been registered in ZooBank, the proposed online registration system for the International Code of Zoological Nomenclature (ICZN). The ZooBank LSIDs (Life Science Identifiers) can be resolved and the associated information viewed through any standard web browser by appending the LSID to the prefix "http://zoobank.org/." The LSIDs for this publication are: zoobank.org/pub:2D9A5324-9B0A-41F5-A66D-803FCE00D43A.

**Reporting summary**. Further information on experimental design is available in the Nature Research Reporting Summary linked to this article.

## Data availability
The authors declare that all data supporting the findings of this study are available within the paper and its supplementary information files.

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

## Acknowledgements

The authors are supported by a Young Thousand Talents Plan of China, a Swedish Research Council Grant (2017-05183) and previously a Marie Curie COFUND Fellowship to T.P.T., a Swedish Research Council Grant to C.B.S. (VR2016-04610) and a grant from Spanish Ministry of Economics, Finance and Competitiveness CGL2017–87631-P to S.C. Financial support from the National Natural Science Foundation of China (41425008, 41621003, and 41720104002 to Z.Z., 41772002 to C.B.S., and 41472015 to G. J.), the Strategic Priority Research Program of the Chinese Academy of Sciences to the Early Life Institute (XDB26000000) and the 111 project (D17013) for the continuous fossil collections of the Xi'an group are sincerely acknowledged. Nobumichi Tamura is thanked for his reconstructions of the taxa illustrated in this paper.

## Author contributions

Z.Z. initiated the collaboration. T.P.T., C.B.S., S.C. and Z.Z. conceived the project. All authors contributed in writing the manuscript and commented on the manuscript at all stages. Z.Z. organized the early fossil campaigns in Hubei. G.J. participated in fieldwork and was responsible for the preparation of specimens. T.P.T., S.C., C.B.S. and G.J. led data analysis and all authors contributed to the interpretation of the results and manuscript preparation.

## Additional information

**Competing interests:** The authors declare no competing interests.

