## [Peer Review File · Nature Communications]

Reviewers' Comments:

Reviewer #1:

Remarks to the Author:

I like this paper, and see no reason to dispute that the three *Yanjiahella* species should be synonymized into one, *Y. biscarpa*, based on arm number. The strength of this paper is the morphological diagnosis, which is very good. *Y. biscarpa* is a fascinating fossil and I am grateful for the authors work on bringing some clarity to its paleontological description. The results presented here are very good. The phylogeny and the analysis of the results however are not good. For these reasons, I recommend that the paper only be published after major revisions. It cannot be published as is for five major reasons: i) nothing is said about the development of the fossil (though this may not be possible), ii) the phylogenetic tree of the deuterostomes used in Figure 3 is very poorly supported, iii) the coding of the matrix is sometimes wrong (comparing an echinoderm exoskeleton with hemichordate tubes), or based on no evidence (e.g., coding for gills and stereome when there are none found). iv) I would also like to see a quantitative analysis of the distal and proximal stalk done to support (or reject) the idea that it lacked a skeleton. v) Finally, the phylogenetic matrix/ tree would benefit from the inclusion of a fossil echinoderm(s) that has a stalk and paired arms (e.g. *Ubaghsicystis* and *Pleurocystites*). The following are my more detailed critiques of the paper. For the most part, they are simple edits to make and I look forward to reading the paper again, so that I may know the evidence based phylogenetic position of the fossil.

Major Revisions

With 35 specimens in hand I wonder if any quantitative measurements can be made to understand the ontogeny of the animal? Does the length vary with width, or the stalk length vary with calyx size, or plate number? This information is critical to compare development of echinoderm fossils with their neontological equivalents. If this is not doable, a simple statement should be made about this matter.

The phylogenetic tree topology (Fig. 3) that puts the pterobranchs as sister to the enteropneust family Harrimanidae is supported only by the 18S gene (Cameron et al. 2000, Halanych et al 2013), which has been problematic with other tree reconstructions including that of the molluscs. The hypothesis that enteropneusts are sister to pterobranchs, on the other hand, is supported by the 5.8S, 16S, and COI genes (Winchell et al. 2002, *Molecular Biology and Evolution*, 19: 762-776), the English, German & French Invertebrate Treatise (Hyman 1959, Horst 1939, Dawydoff 1948), see morphological phylogeny of Cameron (2005), and the most robust molecular phylogenies to date, based on transcriptomic sequence data (Osborn & others, 2011; Cannon & others, 2014). Figure 3 needs to be changed to reflect the enteropneust + pterobranch tree topology on total data, and not one gene. Note, Halanych et al. 2013 used the 18S gene appropriately to identify a deep-sea clade of harrimaniids, and the objective of this paper was not to resolve the enteropneust & pterobranch relationship.

Comments regarding coding of the phylogenetic character matrix, Figure 3.

Fig. 3, character 3 – secretion of tubular exoskeleton is problematic. Only the 505 mya Cambrian acorn worms and pterobranchs are tubicolous. The tubes are fibrous, flexible, and prone to tearing. None are biomineralized, or skeleton, or mucus. Either the authors need to justify their coding hemichordate tubes as homologous with the *Y. biscarpa* exoskeleton (which may be an endoskeleton), or correct the character matrix (Fig. 3, character 3).

All four families of acorn worm, which shared a common ancestor 309 mya (see SI, phylogenetic clock tree in Simakov et al 2015, *Nature*), line their burrows with mucus, and secrete 'tubes' when stressed. These mucous tubes are not homologous to the Cambrian worm tubes because mucus does not

fossilize. Nanglu et al (2016) reply to Halanych et al (2013) and thoroughly rejected the idea that torquatorid tubes are homologous to those of the Cambrian acorn worm fossils. The matrix should be coded such that the secretion of tubes is an ancient hemichordate character lost in modern day enteropneusts, with no equivalent in the echinoderms. This needs to be fixed throughout, including Line 179.

Y. biscarpa appears as the basal group in the authors echinoderm phylogeny. In this scenario, the stalk evolved twice (a second time in the stalked echinoderms) (Fig. 3, character 4). Explain.

The coelomic sacs in hemichordates and echinoderms are organized anterior to posterior as paired protoceols (echinoderm axocoels), mesocoels (echinoderm hydrocoels), and metacoels (echinoderm somatocoels) (Morgan 1891, Gemmill 1914, Gislén 1930, Crawford & Chia 1978). The paired hydrocoeloms are symmetric in extant echinoderm larvae (not the adults) and hemichordates – but for the often left-sided hydropore. The authors need to provide evidence for coding paired, symmetric, hydrocoeloms in *Y. biscarpa* and Ctenocystoids, or change the coding to '?', or change the definition to 'adult echinoderms' (Fig. 3, character 5).

I cannot think of any living animals, that reside in fine sediment, using an anchor, and defecate into that same space. This strikes me as maladaptive. Where do the faeces go? (Fig. 3, character 6). I am OK with the coding of this character – but I think this odd arrangement could be discussed.

(Fig. 3, character 7). The authors found no evidence of gills in *Y. biscarpa*, suggesting that gills are absent, but they instead seem to suggest that they are present, but not seen due to poor preservation. I agree that echinoderms at some point lost their gills because i) when they are found, fossil echinoderm gills are calcified, large, and distinct (sometimes from single fossil specimens), and ii) Simakov et al 2015 Nature, 527: 459–465, doi:10.1038/nature16150, show that gill development genes are in synteny in the seastar *Acanthaster*, suggesting that echinoderm gills are homologous with those of chordates (and hemichordates). Given that gills were not found in 35 specimens, this character should be coded as absent.

Rhabdopleura lacks (ectodermal) gill pores, and *Cephalodiscus* has a pair. Neither *Rhabdopleura* nor *Cephalodiscus* have gill slits (endodermal). Please redo character matrix (Fig. 3, character 7).

Cinctans are symmetric, are they not? Cephalochordates (including the genus *Assymetron*), are quite asymmetric (Fig. 3, character 8). Neither are radial.

Tunicates may be solitary, social, or colonial, but the basal group (appendicularians) are solitary so this coding is good. (Fig. 3, character 9).

The authors find no stereom in *Y. biscarpa* so it should be coded as '?'. This character needs to be checked for the other fossil echinoderms. It is not enough to presume characters are present. I agree with the authors that the stereom microstructure, mouth, gonopores, and (less so) gill slits may be present, but it is entirely misleading to code them as present, when there is no evidence of their form. The presumptive method of coding used in the character matrix gives the appearance that the authors are attempting to shoe-horn *Y. biscarpa* into a preconceived phylogenetic position, rather than one best supported by evidence. It's a fascinating fossil and I want to know what phylogenetic position the data supports (Fig. 3, character 11).

This fossil looks very much like some of the two armed echinoderms in Zamora & Smith (2011). It is not clear to me why it is not phylogenetically nested in their stalked echinoderm phylogeny. Please include a fossil echinoderm(s) that has a stalk and paired arms (e.g. *Ubaghsicystis* and *Pleurocystites*) in the phylogeny.

Zamora & Smith, 2011. Cambrian stalked echinoderms show unexpected plasticity of arm

construction. DOI: 10.1098/rspb.2011.0777

Minor comments

The authors should address the madreporite, from reference 14, in the paper. Is there one, or was the observation incorrect?

Line 41 should read "...mosaic of morphologies that may illustrate..."

Line 120: Why does the absence of a ridge indicate no exoskeleton? Better evidence of different mineral composition using a quantitative technique (e.g., backscatter SEM, elemental analysis) should be done, given the significance of this claim.

Line 120: Cambrian acorn worm fossil are facultative tube dwellers that lack an exoskeleton. This must be corrected throughout the manuscript.

Line 128. Is the muscular organ a reference to the acorn worm proboscis/ pterobranch cephalic shield (protosoma?), because they otherwise have poorly developed muscles.

Line 129. The composition of hemichordate tubes (pterobranch periderm and acorn worm tubes) is not known but may be chiton, keratin, or collagen, and so not necessarily protein. The correct references are 21, 22, 23, 26.

Line 138-140. I don't understand why the rhabdopleurid stolon tube is discussed here when the authors find no homology with it?

Line 159. The anterior paired feeding tentacles would be better called arms, because tentacles are typically paired multiple appendages projecting from the arms. So, in the case, the arms lack tentacles.

Line 184. Cephalodiscus, which is regarded as the sister group to the remaining Pterobranchs (Mitchell et al. 2013) lacks zig-zag fusella.

Mitchell, Melchin, Cameron & Maletz, 2013. Phylogeny of the tube-building Hemichordata reveals that Rhabdopleura is an extant graptolite. *Lethaia*, 46: 34–56.

Line 188. In fact, the Cambrian acorn worm tubes are not interpreted as rigid, but like that of the pterobranch periderm – tough, flexible, tearable and in *Oesia* it is fibrous and interwoven.

Line 209. Reference 9 states that the ancestral deuterostome was a worm, and not a pterobranch. Romer (1967) & Jefferies (1986) put forward the hypothesis that the ancestral deuterostome was a pterobranch. Alternately, you could write that you support the hypothesis of Ref. 9 that rejects the ancestral pterobranch hypothesis.

Reviewer #2:

Remarks to the Author:

The identification of the earliest members of large clades is vital in polarizing characters that help to elucidate phylogenetic relationships. The phylogenetic relationships within echinoderm are particularly difficult to discern such that new data points can be quite important in understanding evolutionary patterns. That being said, I have some concerns regarding this manuscript. I am not sure some of the interpretations of the specimen are warranted given the fossil material. I am not entirely convinced that this animal is an echinoderm. Finally, the reconstruction given as figure 4 could be really quite misleading.

1) Theca. The plates are large and irregular as was reported, but given the preservation there are multiple aspects that we don't know. First, we don't know exactly how these plates are arranged. The reconstruction has three distinctive circlets the second being offset and the third being inline (similar to many crinoids...). I can't see strong evidence of this in the fossils so it doesn't make sense to have

a reconstruction with a very pelmatozoan like thecal structure. Given that these plates are largely the structures that the authors use to put it into the Echinodermata it is troubling that it was found preserved as a mold- the lack of stereom isn't surprising but knowing the mineralogy at very least would be helpful. It is possible that this represents a novel evolution of biomineralization that is unrelated echinoderms. Again, I have no idea regarding the currently specimen but reading through the debate around Echinocrinus would be helpful in seeing how using this character could have issues.

2) Muscular stalk. Superficially this feature was really striking in its resemblance to the stalk in sponges. They also have split proximal and distal stalk with grooves running along the structure that reflect meristic stems. However, this ridge is interpreted as a digestive track rather than a split between skeletal elements, which would result in a periproct medial to the structure which is unlike any echinoderm I am familiar with. Again, their interpretation may be correct, but it is hard to tell with the fossil material.

3) Feeding structures. The feeding structures might be lightly mineralized or lack mineralization at all. This is actually a fairly interesting point-the fact that mineralization is occurring to protect and support the visceral mass rather than to stabilize feeding structures or elevate the animal above the seafloor is interesting in terms of the selective push to biomineralize. One thing to point out is that the number of appendages may not reflect symmetry. Once pentaradial symmetry evolves echinoderms play around with symmetry significantly. Take a look at Sumrall and Wray 2007 in Paleobiology which outlines how developmental changes can build many different symmetry configurations from a starting pentaradial body plan. In fact there are eocrinoids in Cambrian series 3 that have bilateral symmetry with two feeding appendages superimposed on a pentaradial symmetry- see *Ubaghsicystis*. Without a clear view of the oral region it makes it tricky to tell exactly the symmetry of the organism since echinoderms are so plastic in this regard.

4) Implications- The significant finding reported in this paper from an echinoderm perspective is that biomineralization occurred before pentaradial symmetry. The oldest unequivocal echinoderms have tri-radial (helicoptacoids) or pentaradial symmetry (Eocrinoids and edrioasteroids), but most phylogenetic hypotheses place the homalozoans groups in a more basal position. I think given the nature of the Cambrian fossil record it would be interesting to know when biomineralization occurs so we can get more information regarding the animals closer to the node and whether we should be looking at skeletonized or soft bodied fossils. I think the suggestion that biomineralization occurs before the iconic 5-fold symmetry is a widely held idea, but I think we need more fossils and I am not sure if this is the material to convince people.

Response to reviewer's comments

Reviewer #1

I like this paper, and see no reason to dispute that the three *Yanjiahella* species should be synonymized into one, *Y. biscarpa*, based on arm number. The strength of this paper is the morphological diagnosis, which is very good. *Y. biscarpa* is a fascinating fossil and I am grateful for the authors work on bringing some clarity to its paleontological description. The results presented here are very good.

The phylogeny and the analysis of the results however are not good. For these reasons, I recommend that the paper only be published after major revisions. It cannot be published as is for five major reasons: i) nothing is said about the development of the fossil (though this may not be possible), ii) the phylogenetic tree of the deuterostomes used in Figure 3 is very poorly supported, iii) the coding of the matrix is sometimes wrong (comparing an echinoderm exoskeleton with hemichordate tubes), or based on no evidence (e.g., coding for gills and stereome when there are none found). iv) I would also like to see a quantitative analysis of the distal and proximal stalk done to support (or reject) the idea that it lacked a skeleton. v) Finally, the phylogenetic matrix/tree would benefit from the inclusion of a fossil echinoderm(s) that has a stalk and paired arms (e.g. *Ubaghsicystis* and *Pleurocystites*). The following are my more detailed critiques of the paper. For the most part, they are simple edits to make and I look forward to reading the paper again, so that I may know the evidence based phylogenetic position of the fossil.

Reply: The authors thank the reviewer for their thorough and constructive review. We have made substantial changes to address the issues raised and in the process we feel that we have an improved paper. We hope that the reviewer will find some satisfaction with our revised manuscript. Further details are below, however we will briefly discuss the 5 issues raised by reviewer #1 here. 1) We have commented on the difficulty in obtaining an accurate ontogenetic sequence for *Yanjiahella*. 2) We have designed a phylogenetic analysis to test our hypothesis on the placement of *Yanjiahella*. 3) The coding of the characters questioned by the reviewer have been adjusted accordingly. 4) A quantitative analysis of the distal and proximal stalk is problematic, however we have discussed this below. 5) We have included fossil echinoderms in our analysis that have a stalk and 2 arms (both *Ubaghsicystis* and *Pleurocystites* are included).

Major Revisions

1) With 35 specimens in hand I wonder if any quantitative measurements can be made to understand the ontogeny of the animal? Does the length vary with width, or the stalk length vary with calyx size, or plate number? This information is critical to compare development of echinoderm fossils with their neontological equivalents. If this is not doable, a simple statement should be made about this matter.

Reply: A combination of factors made such a quantitative analysis to understand the ontogeny of *Yanjiahella* difficult. Providing accurate measurements was problematic, as the plates of *Yanjiahella* are variable in shape (obscuring reliable plate measurements) and the theca itself is also not consistent in shape (most likely a result of compression of a theca that consists of loosely embedded plates). The distal stalk is also strongly curved and some of the specimens, despite nice preservation are incomplete (such as the specimen in Fig. 2a-c) where the distal stalk is missing. We have however added a sentence in the text to explain this:

'The variable shape of the theca, the strong curvature of the distal stalk and the incomplete preservation of some specimens has hindered any reliable quantitative study and details

regarding the ontogeny of *Y. biscarpa* are unknown.'

2) The phylogenetic tree topology (Fig. 3) that puts the pterobranchs as sister to the enteropneust family Harrimanidae is supported only by the 18S gene (Cameron et al. 2000, Halanych et al 2013), which has been problematic with other tree reconstructions including that of the molluscs. The hypothesis that enteropneusts are sister to pterobranchs, on the other hand, is supported by the 5.8S, 16S, and COI genes (Winchell et al. 2002, *Molecular Biology and Evolution*, 19: 762-776), the English, German & French Invertebrate Treatise (Hyman 1959, Horst 1939, Dawydoff 1948), see morphological phylogeny of Cameron (2005), and the most robust molecular phylogenies to date, based on transcriptomic sequence data (OSBORN & others, 2011; CANNON & others, 2014). Figure 3 needs to be changed to reflect the enteropneust + pterobranch tree topology on total data, and not one gene. Note, Halanych et al. 2013 used the 18S gene appropriately to identify a deep- sea clade of harrimaniids, and the objective of this paper was not to resolve the enteropneust & pterobranch relationship.

Reply: On the suggestions of reviewers and editors we have designed a phylogenetic analysis to test our hypothesis regarding the phylogenetic placement of *Yanjiahella biscarpa*. Consequently the tree topology from the original submission has been replaced by the results of our own analysis.

3) Comments regarding coding of the phylogenetic character matrix, Fig 3.

Fig. 3, character 3 – secretion of tubular exoskeleton is problematic. Only the 505 mya Cambrian acorn worms and pterobranchs are tubicolous. The tubes are fibrous, flexible, and prone to tearing.

None are biomineralized, or skeleton, or mucus. Either the authors need to justify their coding hemichordate tubes as homologous with the *Y. biscarpa* exoskeleton (which may be an endoskeleton), or correct the character matrix (Fig. 3, character 3).

All four families of acorn worm, which shared a common ancestor 309 mya (see SI, phylogenetic clock tree in Simakov et al 2015, *Nature*), line their burrows with mucus, and secrete 'tubes' when stressed. These mucous tubes are not homologous to the Cambrian worm tubes because mucus does not fossilize. Nanglu et al (2016) reply to Halanych et al (2013) and thoroughly rejected the idea that torquatorid tubes are homologous to those of the Cambrian acorn worm fossils. The matrix should be coded such that the secretion of tubes is an ancient hemichordate character lost in modern day enteropneusts, with no equivalent in the echinoderms. This needs to be fixed throughout, including Line 179.

Reply: Although the character matrix from the original submission has changed substantially a number of these characters remain and we have changed the original coding to comply with the reviewers comments.

4) *Y. biscarpa* appears as the basal group in the authors echinoderm phylogeny. In this scenario, the stalk evolved twice (a second time in the stocked echinoderms) (Fig. 3, character 4). Explain.

Reply: In our phylogenetic analysis, *Y. biscarpa* does appear as a stem-group echinoderm which confirms that the posterior appendage evolved numerous times among the echinoderms. We have added a discussion in 'Phylogenetic Significance' to address this scenario. (lines 195- 208).

5) The coelomic sacs in hemichordates and echinoderms are organized anterior to posterior as

paired protoceols (echinoderm axocoels), mesocoels (echinoderm hydrocoels), and metacoels (echinoderm somatocoels) (Morgan 1891, Gemmill 1914, Gislén 1930, Crawford & Chia 1978). The paired hydrocoeloms are symmetric in extant echinoderm larvae (not the adults) and hemichordates – but for the often left-sided hydropore. The authors need to provide evidence for coding paired, symmetric, hydrocoeloms in *Y. biscarpa* and Ctenocystoids, or change the coding to ‘?’, or change the definition to ‘adult echinoderms’ (Fig. 3, character 5).

Reply: In our phylogenetic analysis we have taken the advice of the reviewer and therefore this character state in *Yanjiahella* and ctenocystoids is questioned.

6) I cannot think of any living animals that reside in fine sediment using an anchor, and defecate into that same space. This strikes me as maladaptive. Where do the faeces go? (Fig. 3, character 6). I am OK with the coding of this character – but I think this odd arrangement could be discussed.

Reply: Echinoids have their anus at their upper surface and defecate on themselves and some crinoids (without anal sac or tubes) defecate directly on their oral surface. It seems that echinoderms in general have an interesting relationship with their faeces. To help clarify our view we have discussed in greater detail the possible life habit of *Yanjiahella* in the main text (lines 237-247).

7) (Fig. 3, character 7). The authors found no evidence of gills in *Y. biscarpa*, suggesting that gills are absent, but they instead seem to suggest that they are present, but not seen due to poor preservation. I agree that echinoderms at some point lost their gills because i) when they are found, fossil echinoderm gills are calcified, large, and distinct (sometimes from single fossil specimens), and ii) Simakov et al 2015 *Nature*, 527: 459–465, [doi:10.1038/nature16150](https://doi.org/10.1038/nature16150), shows that gill development genes are in synteny in the seastar *Acanthaster*, suggesting that echinoderm gills are homologous with those of chordates (and hemichordates). Given that gills were not found in 35 specimens, this character should be coded as absent.

Reply: Poor plate preservation does not allow an accurate assessment regarding the presence of gills/pore in *Yanjianella biscarpa*. Thecal plates of *Yanjianella biscarpa* might not have abutted during life so it is possible that gills may have opened between the plates, within the soft tegument as such we have consequently questioned this character state in our analysis.

8) *Rhabdopleura* lacks (ectodermal) gill pores, and *Cephalodiscus* has a pair. Neither *Rhabdopleura* nor *Cephalodiscus* have gill slits (endodermal). Please redo character matrix (Fig. 3, character 7).

Reply: In our phylogenetic analysis we have taken the advice of the reviewer and therefore this character state in the pterobranchs has been clarified.

9) Cinctans are symmetric, are they not? Cephalochordates (including the genus *Assymetron*), are quite asymmetric (Fig. 3, character 8). Neither are radial.

Reply: The body plan of cinctans varies from asymmetrical to nearly bilaterally symmetrical; symmetry is expressed in the shape of the theca and the size/ number of the anterior feeding grooves. The degree of asymmetry in cinctans may relate to the degree of torsion experienced during their metamorphosis from larva to adult. (Rahman and Zamora 2009; Zamora and Rahman 2014). As such we have left cinctans as asymmetrical in our matrix. Cephalochordates are asymmetrical, and this has been changed.

10) The authors find no stereom in *Y. biscarpa* so it should be coded as ‘?’. This character needs to be checked for the other fossil echinoderms. It is not enough to presume characters are present. I agree with the authors that the stereom microstructure, mouth, gonopores, and (less so) gill slits may be present, but it is entirely misleading to code them as present, when there is no evidence of their form. The presumptive method of coding used in the character matrix gives the appearance that the authors are attempting to shoe-horn *Y. biscarpa* into a preconceived phylogenetic position, rather than one best supported by evidence. It’s a fascinating fossil and I want to know what phylogenetic position the data supports (Fig. 3, character 11).

Reply: The reviewer is correct in that we cannot confirm that stereo microstructure is present in *Yanjiahella*. Consequently we have taken the advice of the reviewer and coded this character as ‘?’ in our matrix.

11) This fossil looks very much like some of the two armed echinoderms in Zamora & Smith (2011). It is not clear to me why it is not phylogenetically nested in their stalked echinoderm phylogeny. Please include a fossil echinoderm(s) that has a stalk and paired arms (e.g. *Ubaghsicystis* and *Pleurocystites*) in the phylogeny. Zamora & Smith, 2011. *Cambrian stalked echinoderms show unexpected plasticity of arm construction*. DOI: 10.1098/rspb.2011.0777

Reply: *Yanjiahella* may look similar to two armed echinoderms, such as *Ubaghsicystis* and *Pleurocystites* however the similarities are rather superficial. *Yanjiahella* lacks blastozoan-grade characters such as ambulacra, plated stalk and skeletonized feeding-appendages. These features are present in both derived taxa named by the reviewer. That said we have included both blastozoan genera in our analysis to test if these similarities really are superficial.

Minor comments

12) The authors should address the madreporite, from reference 14, in the paper. Is there one, or was the observation incorrect?

Reply: In the original description (Guo et al. 2012) a structure was interpreted to possibly be a madreporite. We have looked at the specimens and see no structure that would suggest that a madreporite is present. The interpreted structure is not perforated and most likely represents an individual plate.

13) Line 41 should read “...mosaic of morphologies that may illustrate...”

Reply: Corrected

14) Line 120: Why does the absence of a ridge indicate no exoskeleton? Better evidence of different mineral composition using a quantitative technique (e.g., backscatter SEM, elemental analysis) should be done, given the significance of this claim.

Reply: As these fossils are preserved as compressed moulds in siliciclastic sediments the original mineralogy is no longer present. Unlike Burgess Shale-type deposits, such an elemental analysis will only be recording the mineral composition of the host rock. We have undertaken some low vacuum SEM imaging and these have been included in the manuscript (such as Fig. 2e). We have discussed thoroughly in the main text our interpretation regarding the presence of the tubular exoskeleton surrounding the proximal stalk of *Yanjiahella*, based on the differences in flexibility and the absence of transverse ridges that are instead replaced by longitudinal striations

(which are more suggestive of musculature).

15) Line 120: Cambrian acorn worm fossil are facultative tube dwellers that lack an exoskeleton. This must be corrected throughout the manuscript.

Reply: Corrected, we have focused on the fact that the Cambrian enteropneusts live in tubes rather than possess an exoskeleton.

16) Line 128. Is the muscular organ a reference to the acorn worm proboscis/ pterobranch cephalic shield (protosoma?), because they otherwise have poorly developed muscles.

Reply: Yes, it was a reference to the proboscis/cephalic shield.

17) Line 129. The composition of hemichordate tubes (pterobranch periderm and acorn worm tubes) is not known but may be chiton, keratin, or collagen, and so not necessarily protein. The correct references are 21, 22, 23, 26.

Reply: Corrected

18) Line 138-140. I don't understand why the rhabdopleurid stolon tube is discussed here when the authors find no homology with it?

Reply: The authors feel that given the discussion on ambulacraria characters it is important to not just discuss homologous characters.

19) Line 159. The anterior paired feeding tentacles would be better called arms, because tentacles are typically paired multiple appendages projecting from the arms. So, in the case, the arms lack tentacles.

Reply: The authors have not referred to the feeding appendages of *Yanjiahella* as 'tentacles' or 'arms'; instead we refer to them throughout the manuscript as 'appendages'.

20) Line 184. *Cephalodiscus*, which is regarded as the sister group to the remaining Pterobranchs (Mitchell et al. 2013) lacks zig-zag fusella. Mitchell, Melchin, Cameron & Maletz, 2013. Phylogeny of the tube-building Hemichordata reveals that *Rhabdopleura* is an extant graptolite. *Lethaia*, 46: 34–56.

Reply: We have changed the sentence in the text to accommodate this comment. '*Y. biscarpa* lacks a stolon-system, and does not exhibit the unique zig-zag, fusellar structure characteristic of most pterobranchs

21) Line 188. In fact, the Cambrian acorn worm tubes are not interpreted as rigid, but like that of the pterobranch periderm – tough, flexible, tearable and in *Oesia* it is fibrous and interwoven.

Reply: Corrected

Line 209. Reference 9 states that the ancestral deuterostome was a worm, and not a pterobranch. Romer (1967) & Jefferies (1986) put forward the hypothesis that the ancestral deuterostome was a pterobranch. Alternately, you could write that you support the hypothesis of Ref. 9 that rejects the ancestral pterobranch hypothesis.

Reply: We have included the appropriate references as suggested by the reviewer.

Reviewer #2

Comment: The identification of the earliest members of large clades is vital in polarizing characters that help to elucidate phylogenetic relationships. The phylogenetic relationships within echinoderm are particularly difficult to discern such that new data points can be quite important in understanding evolutionary patterns. That being said, I have some concerns regarding this manuscript. I am not sure some of the interpretations of the specimen are warranted given the fossil material. I am not entirely convinced that this animal is an echinoderm. Finally, the reconstruction given as figure 4 could be really quite misleading.

Reply: The authors recognize that the reviewer has some reservations regarding the taxonomic assignment of *Yanjiahella* as a basal echinoderm. Stem group taxa can sometimes be difficult to recognize as they commonly share a mixture of characters that at first glance make the taxa look dissimilar to younger, more derived members of the group. The authors feel that the possession of a plated theca is a significant indicator that we are looking at a basal echinoderm taxon. Despite this hesitation in the above comment, the reviewer continues to compare the morphology of *Yanjiahella* to solute echinoderms and to younger blastozoan taxa, suggestive that similarities between echinoderms and *Yanjiahella* are present.

1) Theca. The plates are large and irregular as was reported, but given the preservation there are multiple aspects that we don't know. First, we don't know exactly how these plates are arranged. The reconstruction has three distinctive circlets the second being offset and the third being inline (similar to many crinoids...). I can't see strong evidence of this in the fossils so it doesn't make sense to have a reconstruction with a very pelmatozoan like thecal structure. Given that these plates are largely the structures that the authors use to put it into the Echinodermata it is troubling that it was found preserved as a mold- the lack of stereom isn't surprising but knowing the mineralogy at very least would be helpful. It is possible that this represents a novel evolution of biomineralization that is unrelated echinoderms. Again, I have no idea regarding the currently specimen but reading through the debate around *Echmatocrinus* would be helpful in seeing how using this character could have issues.

Reply: We have made it clear in the text (Description and Discussion) that the precise plate arrangement is unclear. In regards to the reviewer's comments we have made a slight adjustment to the reconstruction in Figure 4 to show a more irregular plate arrangement and added a further sentence in the figure caption. As *Yanjiahella* specimens are preserved as moulds, despite the fact that we agree with the reviewer 'that knowing the mineralogy would be useful', we are unable to comment on the mineralogy of the plates. Although *Echmatocrinus* was initially considered as a primitive crinoid (Sprinkle, 1973; Sprinkle and Collins, 1998) the echinoderm affinity of the genus has been convincingly refuted (Conway Morris, 1993; Ausich & Babcock, 1998, 2000) and is now thought to be most probably an octocoral (Ausich & Babcock, 1998, 2000). As such we have not included the *Echmatocrinus* in our analysis.

2) Muscular stalk. Superficially this feature was really striking in its resemblance to the stalk in solutes. They also have split proximal and distal stalk with grooves running along the structure that reflect meric stems. However, this ridge is interpreted as a digestive track rather than a split between skeletal elements, which would result in a periproct medial to the structure which is unlike any echinoderm I am familiar with. Again, their interpretation may be correct, but it is hard to tell with the fossil material.

Reply: The authors have discussed the muscular stalk of *Yanjiahella* extensively in the text and we agree with the reviewer that it does not resemble the stalk of more derived echinoderm taxa and it is for this reason (amongst others) that *Yanjiahella* occupies a basal position as a stem-group echinoderm in our analysis. The resemblance with the solute stalk is indeed striking at first glance, however as argued in the text (lines 132-134, 198-208) this is superficial. The solute stem is fully plated, so the longitudinal furrows run along the skeleton. The stalk in *Yanjiahella* is clearly not plated and the distal part of the stalk is not mineralized and depleted of transverse articulations.

3) Feeding structures. The feeding structures might be lightly mineralized or lack mineralization at all. This is actually a fairly interesting point-the fact that mineralization is occurring to protect and support the visceral mass rather than to stabilize feeding structures or elevate the animal above the seafloor is interesting in terms of the selective push to biomineralize. One thing to point out is that the number of appendages may not reflect symmetry. Once pentaradial symmetry evolves echinoderms play around with symmetry significantly. Take a look at Sumrall and Wray 2007 in *Paleobiology* which outlines how developmental changes can build many different symmetry configurations from a starting pentaradial body plan. In fact there are eocrinoids in Cambrian series 3 that have bilateral symmetry with two feeding appendages superimposed on a pentaradial symmetry- see *Ubaghsicystis*. Without a clear view of the oral region it makes it tricky to tell exactly the symmetry of the organism since echinoderms are so plastic in this regard.

Reply: The presence of two appendages projecting from opposing lateral sides of the theca do imprint a distinct bilateral symmetry to *Yanjiahella* and we have interpreted it as such in the manuscript. We have discussed *Ubaghsicystis* in the main text and have included the taxa in our phylogenetic analysis.

4) Implications- The significant finding reported in this paper from an echinoderm perspective is that biomineralization occurred before pentaradial symmetry. The oldest unequivocal echinoderms have tri-radial (helicoplacoids) or pentaradial symmetry (Eocrinoids and edrioasteroids), but most phylogenetic hypotheses place the homalozoans groups in a more basal position. I think given the nature of the Cambrian fossil record it would be interesting to know when biomineralization occurs so we can get more information regarding the animals closer to the node and whether we should be looking at skeletonized or soft bodied fossils. I think the suggestion that biomineralization occurs before the iconic 5-fold symmetry is a widely held idea, but I think we need more fossils and I am not sure if this is the material to convince people.

Reply: The authors agree with the reviewer that *Yanjiahella* confirms that biomineralization occurred before pentaradial symmetry. We agree that this is significant, and we state as much in our abstract (and in the following discussion etc). Although the style of preservation does not allow us to comment in detail regarding the biomineralization processes that governed the formation of the plates in *Yanjiahella*, biomineralization in general was widespread at this stage of the Cambrian (upper Fortunian) (see Kouchinsky et al. 2012). Small shelly fossils representing a variety of groups were already prevalent in carbonate deposits around the globe and it is not unreasonable to think that stem-group echinoderm taxa possessed the ability to biomineralize at this time.

References used in this response:

Ausich, W.I. and Babcock, L.E., 1998. The phylogenetic position of *Echmatocrinus brachiatus*, a probable octocoral from the Burgess Shale. *Palaeontology*, 41(2), pp.193-202.

Ausich, W.I. and Babcock, L.E., 2000. *Echmatocrinus*, a Burgess Shale animal reconsidered. *Lethaia*, 33(2), pp.92-94.

Guo., J., Yong, L., Huiping, H., Qiang, O., Jianren, Z.H.O.U. and Yajuan, Z., 2012. New macroscopic problematic fossil from the early Cambrian Yanjiahe biota, Yichang, Hubei, China. *Acta Geologica Sinica-English Edition*, 86(4), pp.791-798.

Kouchinsky, A., Bengtson, S., Runnegar, B., Skovsted, C., Steiner, M. and Vendrasco, M., 2012. Chronology of early Cambrian biomineralization. *Geological Magazine*, 149(2), pp.221-251.

Morris, S.C., 1993. The fossil record and the early evolution of the Metazoa. *Nature*, 361(6409), p.219.

Rahman, I.A. and Zamora, S., 2009. The oldest cinctan carpoid (stem-group Echinodermata), and the evolution of the water vascular system. *Zoological Journal of the Linnean Society*, 157(2), pp.420-432.

Sprinkle, J. 1973. Morphology and Evolution of Blastozoan Echinoderms. 284 pp. Special Publication of the Museum of Comparative Zoology, Harvard University.

Sprinkle, J. and Collins, D., 1998. Revision of *Echmatocrinus* from the Middle Cambrian Burgess Shale of British Columbia. *Lethaia*, 31(4), pp.269-282.

Zamora, S. and Rahman, I.A., 2014. Deciphering the early evolution of echinoderms with Cambrian fossils. *Palaeontology*, 57(6), pp.1105-1119.

Reviewers' Comments:

Reviewer #1:

Remarks to the Author:

This article is an important one, the description of fossils are good, but the coding of the character matrix is frequently wrong, or unjustified. For these reasons, I recommend that the paper only be published after minor revisions.

I have made several minor edits the manuscript that need to be included, or addressed.

My edits to the character matrix/ character list are on the Supplementary Information file. This needs significant revisions, and new trees constructed.

Reviewer #2:

Remarks to the Author:

I am happy to see this revision and the inclusion of more taxa within the phylogenetic analysis. I think the manuscript is much improved with the extra text. Again, I don't have the expertise in hemichordates as the other reviewer, but from an echinoderm prospective, this animal fits what I would expect to see at the stem of the phylum. I do want to clarify one of my comments, because the point I was making seems to be misinterpreted by the authors.

Regarding *Echmatocrinus*, I agree that no one currently considers this animal to be an echinoderm (the last time I spoke with Jim Sprinkle he agreed it was a coral). Therefore, I was not asking the authors to include that animal into the analysis. My point was broader regarding that scientific discussion. If we have stem group animals that are less than ideally preserved it makes the taxonomic position hard to discern. Basically, there are few characters to connect the stem organisms to any particular group (as is defined by their stem position) and the truly diagnostic characters might be not preserved. This results in the potential focusing on a couple characters that links things together as was done with *Echmatocrinus*. Essentially, the presence of plates and overall vague similarity to crinoids resulted in the placement of that animal into the echinoderms. I think anytime a paper reporting the first representative of any major phylum has the danger of falling into the same trap. As was mentioned in the paper, the current animal falls at the base of the tree and even groups with ctenocystids based on shared absences, which makes sense, but also is tricky given the lack of characters in any stem organism. I think looking at the bootstrap values along the stem reflects the difficulties in placing these types of early organisms.

From looking over the character matrix it largely seems like the grouping is supported by sharing applicable characters and features of the stalk- again this isn't a criticism necessarily, but a byproduct of conducting phylogenetic analysis on organism close to the branch. Therefore, the node linking *Yanjianella* to echinoderms isn't that well supported.

Response to reviewer's comments

Reviewer #1

This article is an important one, the description of fossils are good, but the coding of the character matrix is frequently wrong, or unjustified. For these reasons, I recommend that the paper only be published after minor revisions.

Reply: The authors would like to take this opportunity to thank the reviewer for their extremely helpful comments and suggestions throughout the last two rounds of reviews. We feel this has improved the quality of the paper and we hope that you find satisfaction with the revisions that we have made in accordance with your comments.

I have made several minor edits the manuscript that need to be included, or addressed.

Reply: All the minor edits in the manuscript text have been addressed. We have accepted your suggestions and have made the necessary changes.

My edits to the character matrix/ character list are on the Supplementary Information file. This needs significant revisions, and new trees constructed.

Reply: Please find below our point by point response to your suggestions regarding the character matrix. The character number from the previous review is stated and its new character number is given here in brackets. We have constructed a number of new trees based on your suggestions regarding the coding of particular characters.

1 (deleted). uninformative.

Reply: Character deleted according to reviewer remarks.

2 (now 1). no more evidence of this in *Yanjianella* as other echinoderm fossils - so why the coding difference. Justify or correct.

Reply: Character changed in *Yanjiahella* to absent like other echinoderm fossils.

3 (now 2). There is nothing resembling the neurolated cord of acorn worms in pterobranchs (or nerves that extend to tentacles and stock in acorn worms). Fix.

Reply: Character changed to absent in pterobranchs.

4 (now 3). Cinctans & solutes should code as 0/1

Reply: As explained in the previous reply, cinctans are asymmetrical as are solutes. As such the coding remains as 1/1. We have added a few lines of text under the character in the Supplementary Information to explain our decision to code cinctans as asymmetric (appropriate references are added).

6 (deleted). Do the authors appear to assume this ch. is present in all echinoderm fossils but *Yanjianella*. This should be explained or corrected. Do they mean trisomate? Is so, correct, and explain.

Reply: We changed the coding of *Yanjiahella* to tricoelomate in line with the rest of the echinoderm fossils. This however resulted in all taxa in the analysis (with the exception of the

outgroup) coded as tricoloemate. Consequently, the character was uninformative and we have deleted this character from the matrix.

8 (now 6). Do the authors mean mesosoma? Mesocoels are coelomic cavities - whereas mesosoma is the middle body region. Mesocoels are absent from chordates. Present in hemichordates and living echinoderms. Unknown or presumed in echino. fossils - they should be coded the same. If not, justify the coding.

Reply: The authors used the term mesocoels, based on the reviewer's comments from the first round of reviews (included below). As the reviewer has noted, the mesocoel (hydrocoel in echinoderms) or at least in adult echinoderms are not symmetrical. In extant echinoderms the paired hydrocoeloms are symmetrical in larvae and the larval body plan are also bilaterally symmetrical. The authors have questioned this character in *Yanjiahella* and ctenocystoids as they are the only echinoderm taxa that are bilaterally symmetrical in adult form which may potential indicate that the paired hydrocoeloms are also symmetrical. However, we emended the definition of the character to mention 'as adult' to make it clearer.

Reviewer comment from first round of reviews: 'The coelomic sacs in hemichordates and echinoderms are organized anterior to posterior as paired protocoels (echinoderm axocoels), mesocoels (echinoderm hydrocoels), and metacoels (echinoderm somatocoels) (Morgan 1891, Gemmill 1914, Gislén 1930, Crawford & Chia 1978). The paired hydrocoeloms are symmetric in extant echinoderm larvae (not the adults) and hemichordates – but for the often left-sided hydropore. The authors need to provide evidence for coding paired, symmetric, hydrocoeloms in *Y. biscarpa* and Ctenocystoids, or change the coding to '?', or change the definition to 'adult echinoderms' (Fig. 3, character 5).'

9 (now 7). In the comments to me, developmental data was not possible - but here we have a metamorphic character? The coelomic characters should rather be somatic, because we can see soma in most fossils.

Reply: The Reviewer is correct that reliable developmental data from *Yanjiahella* was not possible (based on morphometrics and due to available material). However, we have coded *Yanjiahella* as not having undergone torsion, as the taxa displays an anterior-posterior digestive tract, which indicates a straight gut. This is unlike more derived Cambrian echinoderms that have experienced torsion that brings their anus into close proximity with their mouth. An anterior-posterior axis has been in previous studies used to suggested that particular early echinoderms (such as solutes) have not undergone torsion (Smith, 2008). We have followed this interpretation and coded all bilateral and asymmetric total group echinoderm taxa as not having undergone torsion. We have developed the character definition to support this coding.

10 (now 8). Perhaps the most canonical ch. of deuterostomes, seen in some Carpoids/ Ctenocystoids/ Cinctans/Stylophorans - but none are coded here. Correct.

See Raman, 2009. - Making sense of carpoids.

Reply: The reviewer is correct in pointing out that some total group echinoderms such as ctenocystoids and cinctans (stylophorans not included in this analysis) are reported as potentially having gill slits. (an opinion we follow, after Rahman and Clausen, 2009). We have followed the reviewer's suggestion and coded both ctenocystoids and cinctans as possessing gill slits and/or pores.

11 (now 9). Also present in echinoderm fossils, but not Yanjianella. The absence of the matrix will be conspicuous to any echinoderm paleontologist. It must be corrected here.

Reply: Up to our knowledge, the presence of tongue bars in “pre-radial” echinoderms has never been mentioned. Indeed, gills in fossil echinoderms would be either evidenced by calcified pores or by the presence of a front chamber interpreted as a pharyngeal basket. In both cases, the exact nature of the slits and the presence of primary and secondary (tongue) bars cannot be assessed (soft-parts). However, the reviewer is right in pointing to the coding here, as the presence of such a character cannot be rejected neither. We have modified it accordingly by questioning this character in *Yanjiahella*, cinctans and ctenocystoids for which gills are or may be present (see new char 8). In addition, this character has been turned to ‘– ‘for all modern taxa without gills.

Reply: 10 (new character). We have added this character according to the reviewer’s suggestion. “Add character: Pharyngeal/ Branchial region. It is an acorn worm synapomorphy, including fossil worms.”: Possession of pharyngeal (branchial) region: no (0); yes (1).

16. Ectodermal skeleton in echinoderm adults. Secreted by mesenchyme cells in pluteus.

Not seen, but presumed, in other fossils here besides *Yanjianella*.

Perhaps instead use i) Calcite ossicles, and ii) stereome as separate characters. Justifiable as it is the canonical echinoderm character.

Reply: All publications checked mention a mesodermal skeleton in echinoderms (Smith, 1990; Nielson, 2012; Arnone et al., 2015; Czarkwiani et al., 2016). However, we have omitted the term to avoid confusion and we have also divided this character into two as suggested by reviewer:

15. (new character according to reviewer suggestion): Possession of ossicles (plates) embedded in the body wall: no (0); yes (1).

16. (modified character according to reviewer’s suggestion): Possession of calcitic skeleton made of stereome: no (0); yes (1).

41 (deleted). All animals produce mucus. Given it does not fossilize, probably best to remove.

Reply: The authors agree with the comment made by the reviewer. We have deleted this character from the matrix.

Reviewer #2

I am happy to see this revision and the inclusion of more taxa within the phylogenetic analysis. I think the manuscript is much improved with the extra text. Again, I don’t have the expertise in hemichordates as the other reviewer, but from an echinoderm perspective, this animal fits what I would expect to see at the stem of the phylum. I do want to clarify one of my comments, because the point I was making seems to be misinterpreted by the authors.

Reply: The authors would also like to take this opportunity to thank the reviewer for their extremely helpful comments and suggestions throughout the last two rounds of reviews. We also feel that the manuscript is much improved with the extra text and phylogenetic analysis. Thank you once again for your input.

Regarding *Echmatocrinus*, I agree that no one currently considers this animal to be an echinoderm (the last time I spoke with Jim Sprinkle he agreed it was a coral). Therefore, I was not asking the authors to include that animal into the analysis. My point was broader regarding that scientific discussion. If we have stem group animals that are less than ideally preserved it

makes the taxonomic position hard to discern. Basically, there are few characters to connect the stem organisms to any particular group (as is defined by their stem position) and the truly diagnostic characters might be not preserved. This results in the potential focusing on a couple characters that links things together as was done with *Echmatocrinus*. Essentially, the presence of plates and overall vague similarity to crinoids resulted in the placement of that animal into the echinoderms. I think anytime a paper reporting the first representative of any major phylum has the danger of falling into the same trap. As was mentioned in the paper, the current animal falls at the base of the tree and even groups with ctenocystids based on shared absences, which makes sense, but also is tricky given the lack of characters in any stem organism. I think looking at the bootstrap values along the stem reflects the difficulties in placing these types of early organisms. From looking over the character matrix it largely seems like the grouping is supported by sharing applicable characters and features of the stalk- again this isn't a criticism necessarily, but a byproduct of conducting phylogenetic analysis on organism close to the branch. Therefore, the node linking *Yanjianella* to echinoderms isn't that well supported.

Reply: The authors agree with the reviewer that it can be tricky to assess stem group taxa as they can show a strange mosaic of characters that may make systematic placement difficult and this also can cause some slight issues when undertaking phylogenetic analyses. It is also probably worth noting that morphological differences between clades that are now distinct (morphologically speaking) may have been relatively minimal in the Cambrian. So, we understand the reviewer's point of view, nobody said dealing with early Cambrian taxa was easy, however the intriguing nature of these creatures easily compensates for any frustration. Very few macrofossils are retrieved from strata of Fortunian age, let alone macrofossils that exhibit discernible morphological characters, which highlights the phylogenetic significance of *Yanjiahella*. The authors feel and hope that we have presented sufficient evidence to show that *Yanjiahella* is morphologically closer to the echinoderms than other members of the ambulacrarians, justifying our results in the manuscript.

Reviewers' Comments:

Reviewer #1:

Remarks to the Author:

I am happy to see the improvements to this paper. Repeating what I have written earlier - this fossil is an important find and the authors have provided a convincing interpretation of its morphology. The matrix is improved and ready for publication as is. My final request is to include a consensus tree (strict, or 50%, or...). I have used McClade, then Paup to run your matrix, via a simple heuristic analysis. Tree length, CI and RI values are as you found, but I found 18 equal length trees and you recovered 12 (Supplementary Fig. 6). This may be due to differences in algorithm and I am not concerned with it. The trees that you present however appear to be the ones that support your Yanjiahella as-basal-echinoderm story.

My 50% Majority-rule consensus of 18 trees puts Yanjiahella in a 4 branch polytomy with Cephalochordates, the Hemichordate clade, Ctenocystoids & the remaining echinoderms. The reason for this is that in 50% of my trees Yanjiahella is more closely related to the hemichordate clade.

I am not disputing your 'basal' hypothesis. It is a good one - but a consensus tree needs to be presented, and then a short explanation of why you chose the trees that you did over the other most parsimonious trees. Although this may weaken the 'basal echinoderm' story, it will strengthen the paper.

Response to reviewer's comments

Reviewer #1

I am happy to see the improvements to this paper. Repeating what I have written earlier - this fossil is an important find and the authors have provided a convincing interpretation of its morphology. The matrix is improved and ready for publication as is. My final request is to include a consensus tree (strict, or 50%, or...). I have used McClade, then Paup to run your matrix, via a simple heuristic analysis. Tree length, CI and RI values are as you found, but I found 18 equal length trees and you recovered 12 (Supplementary Fig. 6). This may be due to differences in algorithm and I am not concerned with it. The trees that you present however appear to be the ones that support your *Yanjiahella* as-basal-echinoderm story.

My 50% Majority-rule consensus of 18 trees puts *Yanjiahella* in a 4 branch polytomy with Cephalochordates, the Hemichordate clade, Ctenocystoids & the remaining echinoderms. The reason for this is that in 50% of my trees *Yanjiahella* is more closely related to the hemichordate clade.

I am not disputing your 'basal' hypothesis. It is a good one - but a consensus tree needs to be presented, and then a short explanation of why you chose the trees that you did over the other most parsimonious trees. Although this may weaken the 'basal echinoderm' story, it will strengthen the paper.

Reply: The authors would like to take this opportunity to thank the reviewer for their extremely helpful comments and suggestions throughout the entire revision process. The authors feel that your detailed knowledge of the ambulacrarians has improved the quality of the paper and we are grateful for the time and effort that you have put into this. You are correct, that a 50% majority-rule consensus tree via a simple heuristic analysis resolves *Yanjiahella* in a closer relationship with the hemichordates rather than the echinoderms. We have included the 50% majority-rule consensus tree that we obtained under heuristic methods in the Supplementary Information (Supplementary Figure 7a) as requested. We have also included another two trees, a 75% majority-rule consensus tree (obtained using heuristic methods) and a strict consensus tree (obtained using New Technology Search) to show a greater range of results obtainable depending on the analysis that you utilise. The 75% majority-rule consensus tree (obtained using heuristic methods) instead shows *Yanjiahella*, ctenocystoids and cinctans as unresolved basal ambulacrarians (Supplementary Figure 7b) and the strict consensus tree (obtained using New Technology Search) shows *Yanjiahella* once again as a basal echinoderm (Supplementary Figure 9). The ambulacrarians are a morphologically disparate clade and as shown in our bootstrap values, the base of the clade is relatively poorly constrained. This is most likely a combination of this morphological diversity seen across the clade but also the fact that morphological differences between clades that are now distinct (morphologically speaking) may have been relatively minimal in the earliest Cambrian. But we hope that the addition of new trees that we present here has strengthened the paper and at the same time we don't think that this has distracted from the 'basal echinoderm' story. The authors feel that we have in addition to the phylogenetic analyses, presented sufficient evidence from the fossils themselves to show that *Yanjiahella* is

morphologically closer to the echinoderms than other members of the ambulacrarians, justifying our results in the manuscript.